# SyncKV: A Syncopated Scheduling Approach to KV Cache Compression for Efficient Long-Context LLM Inference

## Abstract

KV cache accelerates the inference of Large Language Models (LLMs) by caching the key and value states of previous tokens, but its linearly increasing memory footprint poses a huge bottleneck for long-context tasks. To mitigate this, many previous studies evict unimportant tokens based on attention scores from the pre-fill stage or cumulative attention. However, by permanently evicting tokens, such static compression algorithms fail to preserve globally important tokens, as they overlook the "attention drift" phenomenon inherent in inference. Our analysis highlights this drift, showing that after generating just 50 tokens, the set of important tokens retains only about a 30% overlap with the one identified during the prefill stage. To address this, our core innovative insight is twofold: (1) the set of important tokens exhibits high temporal locality across adjacent generation steps, and (2) this set is highly similar among attention heads within the same layer. Based on these insights, we propose SyncKV, a training-free dynamic KV cache compression method. SyncKV takes advantage of these properties through a novel syncopated strategy in which a few "representative heads" periodically identify important tokens, triggering an asynchronous upload of the relevant KV cache from the CPU. We designed a parallelization strategy that overlaps the I/O overhead with the subsequent forward computing stage, thereby effectively hiding the delay of data transmission and achieving an acceleration effect. Experiments show that SyncKV has achieved state-of-the-art performance in multiple long-context benchmarks, reducing the GPU memory usage of the KV cache by up to 80%. Our code will be open-source.

## 1 Introduction

The landscape of artificial intelligence is currently being reshaped by the wave of LLMs, whose powerful capabilities have been validated in numerous interactive applications, such as conversational assistants capable of complex dialogues (OpenAI et al., 2024; Team et al., 2025) and autonomous agents capable of planning and performing tasks (Wu et al., 2023; Park et al., 2023). The success of these applications converges on a core requirement: models must possess the ability to process and retain long-form contextual information. To push beyond the context limits of existing models, some work has proposed various innovative architectures with significant success (Dao et al., 2022; Chen et al., 2024; Grattafiori et al., 2024). However, all models based on the Transformer architecture (Vaswani et al., 2017) rely on an optimization mechanism known as the KV cache during auto-regressive generation. Although this mechanism avoids redundant computations by storing the intermediate states of all previous tokens, it comes at the cost of substantial GPU memory consumption that grows linearly with sequence length. This makes the KV cache the primary obstacle to deploying long-context LLMs on resource-constrained hardware. The work on Minference (Jiang et al., 2024) highlights the inherent sparsity of the attention mechanism in long contexts, showing that just 4k tokens out of 128k can account for 96.4% of the total attention weight. Consequently, a major line of research has focused on evicting entire unimportant token entries from the cache. These methods typically leverage various heuristics based on attention scores to identify and retain a critical subset of tokens from the initial prompt (Xiao et al., 2023; Zhang et al., 2023; Li et al.,

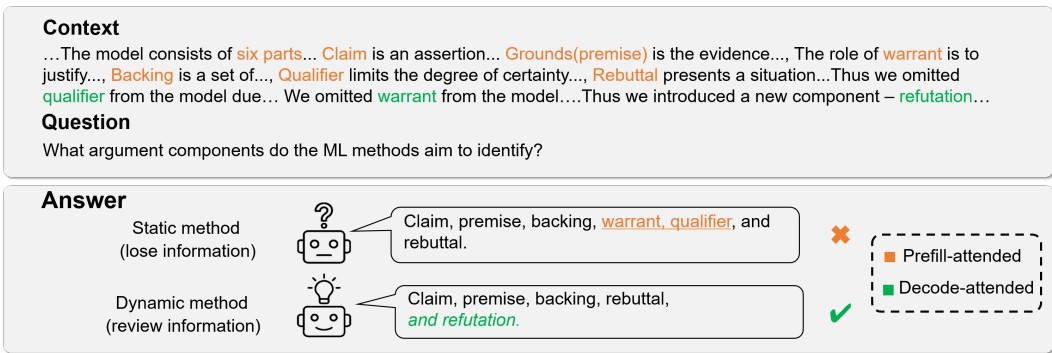

Figure 1: A comparative example. The static compression method only focuses on the important tokens (orange words) in the prefill stage, thereby losing information, while the dynamic method can review the less important information (green words) in the decode stage.

2024), or maximize information density by allocating different KV cache budgets to different layers or heads (Cai et al., 2024).

We argue that such methods, which rely on static decisions based on historical scores, carry a fundamental risk. In long dialogues or complex reasoning, the focus of attention is dynamic, a phenomenon we call attention drift. As illustrated in Figure 1, a detail that appears unimportant in the early stages may become critical to resolving the task later on. Once a static eviction policy makes an incorrect eviction decision, the information loss is irreversible, thus limiting the model's deep reasoning capabilities.

To overcome this, we propose a dynamic approach that avoids premature information loss. Our work is built upon two key observations about the attention mechanism's behavior: (1) temporal locality, The set of most-attended-to tokens exhibits high temporal locality, with significant overlap between adjacent decoding steps, and (2) intralayer similarity, this set is highly similar among many attention heads within the same layer. Based on these insights, we introduce SyncKV, a novel dynamic KV cache compression framework that leverages these spatio-temporal properties of attention. SyncKV employs a syncopated strategy, where a few "representative heads" are first selected through a one-time offline clustering. During the initialization phase of inference, the vast majority of the KV cache is offloaded to CPU memory, with only a small, critical subset retained on the GPU. Subsequently, in the decode stage, these representative heads periodically access a wider context to dynamically identify the current set of most important tokens, which triggers an asynchronous upload of the corresponding cache slices from the CPU to the GPU. In the intervening steps, all non-representative heads perform efficient computation on this newly fetched, highly sparse cache subset. This approach ensures that only the representative heads bear the periodic retrieval cost, while the bulk of computation operates on a highly compressed cache, drastically reducing GPU memory overhead with negligible impact on model performance.

Our extensive experiments were conducted on mainstream large language models, including the Llama and Qwen series, and evaluated under various compression rates on multiple long-context benchmarks, such as LongBench (Bai et al., 2023), SCBench (Li et al., 2025), and more. The results demonstrate that, compared to other advanced compression algorithms, SyncKV achieves state-of-the-art performance on a variety of tasks.

## 2 RELATED WORK

### 2.1 KV CACHE EVICTION

Existing work has demonstrated the inherent sparsity of the attention mechanism in long contexts (Jiang et al., 2024). Therefore, many KV cache evicting works are studying how to effectively retain the important tokens in the inference stage and evict the unimportant ones. StreamingLLM (Xiao et al., 2023) and LM-Infinite (Han et al., 2023) relied on a simple positional heuristic to retain initial and final tokens. This simple eviction strategy has lost a large amount of information. To address

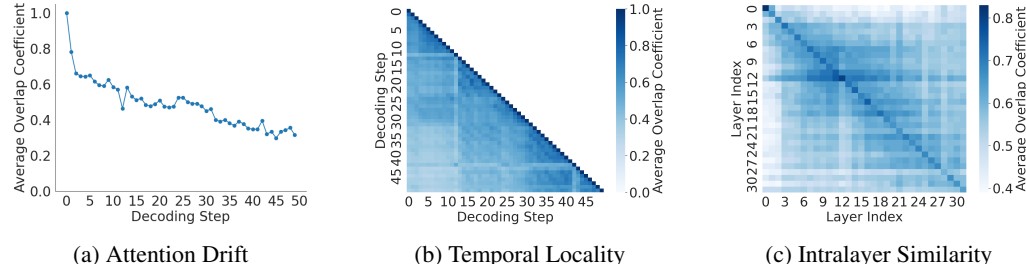

Figure 2: Visualization of key observations. (a) Attention Drift: The average overlap between current and initial steps decreases as decoding progresses. (b) Temporal Locality: The heatmap's darker diagonal indicates high attention overlap between adjacent steps. (c) Intralayer Similarity: The darker main diagonal shows that attention heads are much more similar within a layer than between layers.

this, subsequent methods introduced more sophisticated, attention-driven indicators to identify less important tokens. H2O (Zhang et al., 2023) uses cumulative attention to eliminate the last token. RoCo (Ren & Zhu, 2024) uses mean attention scores to assess token importance to construct a robust scope of eviction. SnapKV (Li et al., 2024) uses window attention clustering in the prefill stage to obtain the set of important tokens. CAKE (Qin et al., 2025) introduces a cascading and adaptive strategy that assesses layer-specific preferences to rationally distribute cache resources. However, all the methods mentioned above have a fatal problem. Tokens that are regarded irrelevant and evicted at a certain stage cannot be retrieved. They may become important in subsequent processes, resulting in significant information loss. Some works have recognized the importance of dynamic selection, such as Quest (Tang et al., 2024). However, it has not led to a decrease in memory usage and has lost accuracy.

## 2.2 OTHER KV CACHE COMPRESSION METHODS

Beyond token-level schemes, recent studies have pursued KV cache compression from several complementary perspectives. DuoAttention (Xiao et al., 2024) only retains the full KV cache for the retrieval attention head and uses a constant length cache for the remaining heads. ThinK (Xu et al., 2024) prunes the least significant key cache channels based on a query-dependent interaction score. MiniCache (Liu et al., 2024a) achieves hierarchical compression by merging highly similar KV cache of adjacent layers from the middle section to the deep section of the model. Some methods also attempt to quantize the KV cache, such as Kvquant (Hooper et al., 2024), Kivi (Liu et al., 2024b), etc. These quantization methods are orthogonal to the token-level eviction strategy of KV cache, and can be combined to further substantially reduce GPU memory overhead.

## 3 OBSERVATION

The effectiveness of our proposed algorithm, which will be detailed in this chapter, is based on several fundamental observations regarding the behavior of the attention mechanism during autoregressive generation. These observations motivate a novel strategy for compressing the KV cache.

## 3.1 ATTENTION DRIFT

Numerous studies on KV cache compression (Li et al., 2024; Cai et al., 2024; Qin et al., 2025) confirm the effectiveness of retaining tokens with top $k$ window attention scores during the prefill stage. However, we find that this set of high-attention tokens is dynamic. To quantify this phenomenon, we use the overlap coefficient (McGill, 1979) to measure the similarity between the sets of top $k$ tokens from any two sources, $\mathcal{T}_k(X)$ and $\mathcal{T}_k(Y)$, calculated as follows:

$$\text{Overlap}(\mathcal{T}_k(X), \mathcal{T}_k(Y)) = \frac{|\mathcal{T}_k(X) \cap \mathcal{T}_k(Y)|}{\min(|\mathcal{T}_k(X)|, |\mathcal{T}_k(Y)|)}, \quad (1)$$

where $X$ and $Y$ are the attention scores from two sources (e.g., different time steps or different attention heads) and $\mathcal{T}_k(\cdot)$ is the set of indices of the top $k$ tokens from a given score. As illustrated in Figure 2 (a), our analysis reveals that as the generation process progresses, the overlap between the set of high-attention tokens for the prefill and current steps continuously decreases, with the overlap coefficient dropping to 30% after only 50 decoding steps. This indicates that tokens identified as important at one point quickly become less relevant, and relying on a fixed, historical set of important tokens can degrade the model's performance on long-sequence generation. Therefore, SyncKV avoids directly evicting tokens to cause information loss. Instead, it offloads the temporarily unimportant KV cache to the CPU memory for subsequent retrieval.

## 3.2 Temporal Locality

Based on attention drift, we discovered another phenomenon: high-attention tokens have a high overlap coefficient in adjacent decoding steps. We define it as temporal locality. As shown in Figure 2 (b), we calculated the average overlap coefficient across all attention heads for the sets of high-attention tokens between each decoding step and found that the overlap for adjacent steps remains at a consistently high level. This indicates a strong temporal locality in the attention mechanism's focus. It suggests that while the set of important tokens does evolve over the long term, its composition changes gradually and predictably in the short term. This finding is significant for cache design, as it implies that the most recently identified high-attention tokens are ideal candidates for retention, given their high probability of remaining important in the immediate future. Using the principle of temporal locality, SyncKV performs a dynamic evaluation periodically, every $m$ steps. In the intervening steps, attention is computed by reusing the set of high-attention tokens identified at the most recent evaluation point. This periodic evaluation strategy is highly effective because temporal locality ensures this token set remains a high-fidelity proxy for several subsequent steps.

## 3.3 Intralayer Similarity

In addition to the temporal dynamics of attention, we also investigated spatial redundancy across attention heads. To quantify intralayer relationships, we define layer-level similarity as the average of the overlap coefficients from each head in a source layer to a target layer. The result, shown in Figure 2 (c), reveals a strong intralayer similarity, which is significantly higher than the interlayer similarity. This high degree of intralayer redundancy suggests that not all heads contribute unique information; many are functionally similar. SyncKV is designed to directly exploit this redundancy. It employs K-means clustering to group functionally similar heads within each layer and designates a single representative head for each cluster. This representative head is responsible for tracking the attention dynamics for the entire cluster, while the other non-representative heads simply synchronize with it. By treating a group of similar heads as a single unit, SyncKV drastically reduces computational and GPU memory overhead without a significant loss of information.

## 4 Method

We propose SyncKV, an efficient KV cache compression framework for long-context LLM inference. The core idea of SyncKV originates from two key insights: temporal locality and intralayer similarity. Based on these insights, this section will introduce the core components of SyncKV in order: Offline Head Clustering, Initialization, and the two alternating decode phases, Anticipation and Suspension. The pseudocode for SyncKV is detailed in Appendix A, and its general workflow is illustrated in Figure 3.

## 4.1 Offline Head Clustering

To leverage the model's intralayer similarity, we introduce a offline analysis phase. We first compute the similarity $S_{i,j}^{(l)}$ between any two heads of attention, $h_i$ and $h_j$, within each layer $l$. This similarity is measured by the average overlap coefficient of their sets of top $k$ attention indices over a representative corpus and multiple time steps $t$:

$$S_{i,j}^{(l)} = \mathbb{E}_{t,\text{data}}\left[\text{Overlap}(\mathcal{T}_k(A_t^{(l,i)}), \mathcal{T}_k(A_t^{(l,j)}))\right],\tag{2}$$

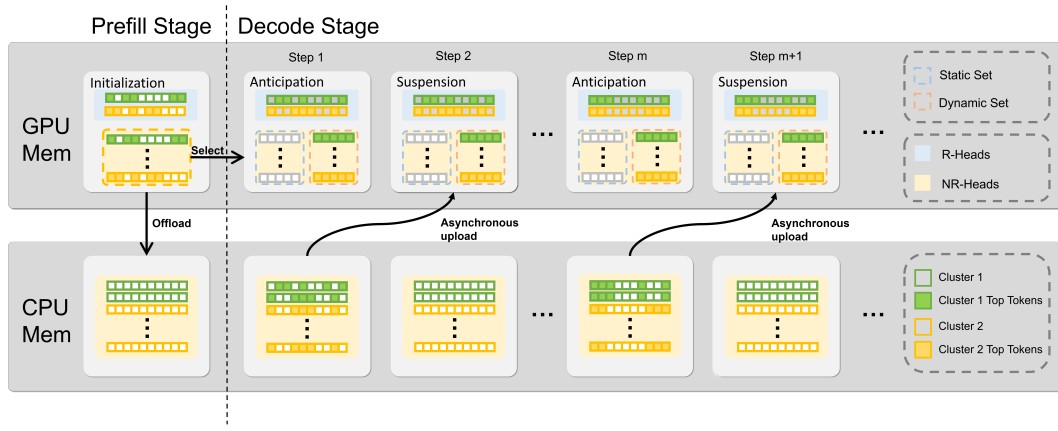

Figure 3: The workflow of SyncKV. After the initialization step in the prefill stage, the decode stage cycles between two steps: (1) the anticipation step, where R-Heads select top $k_{dyn}$ indices to trigger an asynchronous upload from the CPU; and (2) the suspension step, where all heads compute by reusing the existing sparse cache on the GPU.

where $A_t^{(l,i)}$ and $A_t^{(l,j)}$ is the attention score for head $h_i$ and $h_j$ in layer $l$ at time step $t$, and $k$ is the predefined top $k$ value. Based on this similarity matrix, we apply K-Means clustering to partition heads into distinct clusters, $\mathcal{C}_g$. In each cluster, the head closest to the cluster centroid is selected as the Representative Head (R-Head), and all other heads are treated as Non-Representative Heads (NR-Heads). The details of our offline head clustering process, including robustness analysis, are presented in Appendix B.1.

This clustering establishes an efficient "leader-follower" working model. During inference, the R-Heads undertake the more computationally expensive context-aware task. They are responsible for drawing attention over a broader context at critical moments to dynamically identify the most important set of top $k$ tokens. Meanwhile, NR-Heads directly reuse the top $k$ indices calculated by the R-Head of their cluster. In this way, the attentional focus of the entire cluster is unified, and the computation for the NR-Heads is drastically simplified, as they only need to perform attention on the highly sparse KV cache subset pre-selected by the R-Heads.

## 4.2 INITIALIZATION: SELECTING AND OFFLOADING CACHE

SyncKV inference process begins with the prefill stage, which aims to efficiently process the prompt and establish the initial context state for the subsequent decode stage. During this stage, the model first processes the entire input sequence on the GPU to generate the complete initial KV cache. Immediately following this, the KV cache for NR-Heads is asynchronously offloaded to pinned CPU memory, establishing a "context bank" for efficient future access.

To facilitate the first step of the decode stage, an initial set of top $k$ indices is determined by performing attention on the last token of the input sequence. This approach is motivated by the findings of SnapKV (Li et al., 2024), which demonstrated that attention scores computed over a local window of recent tokens serve as a strong predictor for identifying tokens that will be important in subsequent decoding steps:

$$\mathcal{I}_0^{(g)} = \mathcal{T}_k\left(A_{L-1}^{(l,h_r)}\right), \tag{3}$$

where $\mathcal{I}_0^{(g)}$ is the initial set of top $k$ indices for group $g$, and $A_{L-1}$ represents the attention scores calculated at the last position of the input sequence in layer $l$ by the Representative Head $h_r$.

To strike a balance between the global information budget in the prefill stage and the attention drift budget in the decode stage, we introduced the static ratio $\rho$. It divides $\mathcal{I}_0^{(g)}$ into a static part and a dynamic part:

$$\mathcal{I}_0^{(g)} = \mathcal{I}_S^{(g)} \cup \mathcal{I}_{D,0}^{(g)}, \tag{4}$$

where $\mathcal{I}_S^{(g)}$ is the static set, which comprises a fraction of the most important tokens of $\mathcal{I}_0^{(g)}$ corresponding to the ratio $\rho$, and its corresponding KV cache is permanently retained on the GPU throughout the decoding process; $\mathcal{I}_{D,0}^{(g)}$ is the initial dynamic set, defined by the remaining budget, a ratio of $1 - \rho$, which will be populated during the anticipation steps with newly important tokens uploaded from the CPU. This strategy prevents the model from forgetting critical global information by using the static cache, while enabling it to flexibly retrieve important, timely content through the dynamic cache. At the end of this stage, in GPU memory, the R-Heads retain their complete KV cache, while the NR-Heads retain the initial $\mathcal{I}_0^{(g)}$ KV cache.

### 4.3    ANTICIPATION: UPDATING THE DYNAMIC CACHE

The anticipation step serves as a proactive dynamic sparse context retrieval mechanism, initiating a computational cycle defined by the synchronization stride $m$. During an anticipation step, which occurs once every $m$ steps, R-Heads perform an attention calculation to produce an accurate attention score vector that reflects the current focus of inference:

$$A_t^{(l,h_r)} = \mathrm{Softmax} \left( \frac{q_t^{(l,h_r)} \left( K_{\mathrm{GPU}}^{(l,h_r)} \right)^T}{\sqrt{d}} \right), \tag{5}$$

where $A_t^{(l,h_r)}$ is the attention score computed at step $t$ in layer $l$ by the R-Head $h_r$; $q_t^{(l,h_r)}$ is the current query vector; $K_{\mathrm{GPU}}^{(l,h_r)}$ is the corresponding key cache for R-head stored on the GPU; and $d$ is the dimension of the key vectors. Based on this score vector, a new set of top $k$ indices, $\mathcal{I}_{D,t+1}^{(g)}$ is identified for the dynamic cache:

$$\mathcal{I}_{D,t+1}^{(g)} = \mathcal{T}_{k_{dyn}} \left( A_t^{(l,h_r)} \right), \tag{6}$$

where the dynamic budget is set to $k_{dyn} = \lfloor (1 - \rho)k \rfloor$. This new set subsequently initiates an asynchronous data update for the entire cluster $\mathcal{C}_g$. The corresponding KV cache slices within the cluster are fetched from the global context on the CPU and uploaded to the GPU:

$$\begin{aligned} K_{D,t+1}^{(l,h \in \mathcal{C}_g)} &\leftarrow K_{\mathrm{CPU}}^{(l,h \in \mathcal{C}_g)} \big[ \mathcal{I}_{D,t+1}^{(g)} \big], \\ V_{D,t+1}^{(l,h \in \mathcal{C}_g)} &\leftarrow V_{\mathrm{CPU}}^{(l,h \in \mathcal{C}_g)} \big[ \mathcal{I}_{D,t+1}^{(g)} \big]. \end{aligned} \tag{7}$$

This step is termed anticipation because it proactively identifies and fetches the critical KV cache subset by tracking the attention drift, which helps in recovering the potential accuracy loss. This mechanism is crucial, as by periodically executing the anticipation step, the model can dynamically reevaluate and retrieve older information that has become important in the current phase, effectively preventing accuracy degradation caused by information loss. This ensures that the model always has access to the most relevant current context during long-context generation.

### 4.4    SUSPENSION: QUERYING THE SPARSE CACHE

Following each anticipation step, the system enters a series of suspension steps. This stage is defined by no extra overhead computation, designed to fully capitalize on the sparse context prepared by the anticipation step to maximize token generation throughput.

Critically, the NR-Heads attention mechanism no longer operates in the full context. Instead, they jointly attend to a KV cache subset composed of the static set and the dynamic set, where the latter was uniformly loaded for their cluster during the last anticipation step, denoted $t_A$. For any suspension step $t$ in the interval $(t_A, t_A + m]$, the KV cache used for the calculation remains constant:

$$\begin{aligned} K_{D,t+1}^{(l,h)} &= K_{D,t}^{(l,h)}, \\ V_{D,t+1}^{(l,h)} &= V_{D,t}^{(l,h)}. \end{aligned} \tag{8}$$

This step is termed suspension because NR-heads sustain and reuse the sparse $\mathcal{I}_t^{(g)}$ KV cache subset, obtained during the previous anticipation step, for a series of subsequent decoding steps. This sustained context approach creates an I/O-free computational phase, which in turn significantly reduces the complexity of the attention mechanism and directly translates to a substantial acceleration in token generation throughput.

Table 1: The performance of single-step inference on LongBench (Bai et al., 2023). *Italics* indicate that the model uses a full attention baseline. **Bold** indicates the best performance under the same model. Detailed results are presented in Appendix C.3

| Methods | Mem% | Avg. | Single-Doc QA | Multi-Doc QA | Summarize | Few-Shot | Synthetic | Code |
|---|---|---|---|---|---|---|---|---|
| *Llama-3.1-8B-Instruct* | 100% | 49.76 | 43.39 | 42.76 | 29.20 | 69.39 | 51.81 | 62.02 |
| Quest | 100% | 46.73 | 42.05 | 37.38 | 28.12 | 64.81 | **52.98** | 55.03 |
| StreamingLLM | 50% | 38.97 | 27.98 | 32.93 | 22.18 | 62.85 | 35.38 | 52.51 |
| H2O | 50% | 41.82 | 37.48 | 37.44 | 27.36 | 67.55 | 32.59 | 48.52 |
| SnapKV | 50% | 44.51 | 37.25 | 38.29 | 24.63 | 68.22 | 47.88 | 50.81 |
| CAKE | 50% | 48.93 | 43.13 | 42.37 | 27.83 | **68.94** | 52.09 | 59.23 |
| SyncKV | 50% | **49.07** | **43.57** | **42.56** | **28.27** | 68.62 | 52.08 | **59.34** |
| *Qwen2.5-7B-Instruct* | 100% | 49.70 | 41.59 | 43.96 | 26.72 | 67.56 | 54.75 | 63.63 |
| Quest | 100% | 46.66 | 40.14 | 41.73 | 22.95 | **67.75** | 51.25 | 56.13 |
| StreamingLLM | 30% | 41.40 | 31.08 | 36.90 | 23.27 | 65.21 | 33.00 | 58.91 |
| H2O | 30% | 38.54 | 32.45 | 36.66 | **24.03** | 59.87 | 37.50 | 40.71 |
| SnapKV | 30% | 43.91 | 34.84 | 40.31 | 20.54 | 65.31 | **54.75** | 47.72 |
| CAKE | 30% | 48.09 | 40.74 | 42.95 | 23.69 | 66.77 | 54.25 | **60.10** |
| SyncKV | 30% | **48.18** | **40.84** | **44.43** | 22.63 | **67.75** | 54.25 | 59.17 |
| *Qwen2.5-14B-Instruct* | 100% | 49.86 | 42.50 | 51.57 | 23.81 | 70.50 | 52.80 | 57.99 |
| Quest | 100% | 46.48 | 39.68 | 47.96 | 20.33 | 67.36 | 50.93 | 52.62 |
| StreamingLLM | 30% | 39.69 | 26.71 | 40.89 | 20.85 | 66.77 | 27.18 | **55.75** |
| H2O | 30% | 37.57 | 30.28 | 43.28 | 22.07 | 64.45 | 23.08 | 42.27 |
| SnapKV | 30% | 44.17 | 28.90 | 48.53 | 19.48 | 68.79 | 50.38 | 48.92 |
| CAKE | 30% | 48.20 | 40.98 | 50.83 | **22.75** | 69.56 | 51.23 | 53.84 |
| SyncKV | 30% | **48.55** | **41.23** | **51.03** | 21.05 | **70.68** | **52.15** | 55.13 |
| *Llama-3.1-70B-Instruct* | 100% | 53.10 | 43.44 | 50.88 | 28.48 | 70.38 | 55.00 | 70.39 |
| Quest | 100% | 50.76 | 43.13 | 50.11 | 26.55 | 68.17 | 54.25 | 62.36 |
| StreamingLLM | 20% | 44.12 | 30.73 | 44.86 | 17.44 | 56.80 | 54.25 | 60.63 |
| H2O | 20% | 44.80 | 36.60 | 46.35 | 17.32 | 61.20 | 53.35 | 54.00 |
| SnapKV | 20% | 48.28 | 38.01 | 50.07 | 19.62 | 68.57 | 54.25 | 59.16 |
| CAKE | 20% | 51.83 | 43.03 | 50.13 | **27.20** | 68.85 | 54.75 | 67.01 |
| SyncKV | 20% | **51.92** | **43.21** | **50.58** | 26.00 | **69.36** | 54.75 | **67.63** |

## 4.5 Parallelized Data Transfer

In dynamic caching strategies that move data between the CPU and GPU, a core challenge is the significant I/O overhead of this two-way transfer. If not handled properly, this latency can completely nullify the acceleration gains from memory savings. SyncKV addresses this challenge through a carefully designed parallelization strategy.

In the prefill stage, SyncKV uses asynchronous transfer to hide the I/O latency by overlapping it with the attention computation. In the decode stage, the core of this mechanism lies in the asynchronous coordination between the anticipation and suspension steps. When an R-Head identifies the new set of important tokens, it immediately triggers an asynchronous upload command from the CPU to the GPU. Consequently, this time-consuming data transfer occurs in the background, with its latency being overlapped and hidden by the computation of both the current anticipation step and the subsequent suspension steps. SyncKV hides I/O latency by parallelizing computation and data transmission, and its efficient sparse attention computation is sufficient to offset additional overhead, thereby jointly enhancing throughput.

## 5 Experiments

### 5.1 Experiment Settings

**Backbone LLMs.** We selected multiple representative open-source LLMs for our experiments: Llama-3.1-8B-Instruct, Qwen2.5-7B-Instruct, Qwen2.5-14B-Instruct and Llama-3.1-70B-Instruct. All of these models support a context length of up to 128K.

**Implementation.** All performance and latency experiments were conducted on a single NVIDIA A100 GPU and a 32-core Intel Xeon Gold 6326 processor at 2.90GHz. To demonstrate the or-

Table 2: Performance comparison of SyncKV and baselines on multi-step reasoning tasks from SCBench. All methods were evaluated at 30% Mem%.

| Methods | Avg. | Sem. Retr. | Global Info. | Multi-Task |
|---|---|---|---|---|
| *Llama-3.1-8B-Ins.* | 44.37 | 40.66 | 34.01 | 58.45 |
| StreamingLLM | 26.82 | 28.47 | 33.27 | 18.72 |
| H2O | 23.25 | 25.59 | 30.20 | 13.95 |
| SnapKV | 29.24 | 24.58 | **35.32** | 27.83 |
| CAKE | 34.11 | 32.11 | 34.52 | 35.70 |
| SyncKV | **37.59** | **33.64** | 34.47 | **44.66** |
| *Qwen2.5-7B-Ins.* | 25.93 | 21.80 | 33.64 | 22.36 |
| StreamingLLM | 20.68 | 11.28 | **35.65** | 15.10 |
| H2O | 17.62 | 13.03 | 32.05 | 7.77 |
| SnapKV | 19.75 | 14.77 | 34.16 | 10.33 |
| CAKE | 23.55 | 20.32 | 32.10 | **18.22** |
| SyncKV | **24.66** | **21.69** | 34.30 | 18.00 |

Table 3: Ablation study of the core components of SyncKV. Note that "Prefill" and "Decode" refer to latency in seconds (s).

| Methods | Mem% | Avg. | Prefill | Decode |
|---|---|---|---|---|
| SyncKV | **50%** | 48.12 | 2.45 | **0.04** |
| w/o Init. | 100% | 48.12 | **2.19** | 0.05 |
| w/o Anticip. | 50% | 47.75 | 2.43 | **0.04** |
| w/o Susp. | **50%** | **48.50** | 2.63 | 0.21 |
| w/ $N_r$=1 | 50% | 47.86 | 2.45 | **0.04** |
| w/ $N_r$=2 | 50% | **48.12** | **2.42** | **0.04** |
| w/ $N_r$=3 | 50% | 47.63 | 2.52 | 0.07 |
| w/ $m$=1 | 50% | **48.50** | 2.63 | 0.21 |
| w/ $m$=5 | 50% | 48.12 | **2.45** | **0.04** |
| w/ $m$=10 | 50% | 47.62 | 2.51 | **0.04** |

thogonality of SyncKV with quantization, we applied 4-bit weight quantization to all models with bitsandbytes (Dettmers et al., 2021). To ensure a fair comparison, we standardized the evaluation criteria by requiring all algorithms to retain a prefetched KV cache equivalent to a fixed percentage of GPU memory (Mem%), rather than a fixed token budget. For the hyperparameter configuration, we set $N_r$ to 2 for the Llama model and 1 for Qwen model. The synchronization stride $m$ was set at 5 for both models and the static ratio $\rho$ was set to 0.5. Finally, to evaluate performance under different budget constraints, we set Mem% to 20%, 30%, and 50%. Further details of our experimental setup are presented in the Appendix C.

## 5.2 PERFORMANCE ON VARIOUS TASKS

**Single-Step Reasoning Performance.** To comprehensively evaluate SyncKV's single-step inference performance, we conducted extensive experiments on LongBench, with detailed results summarized in Table 1. For instance, even with a significant memory reduction on the Llama-3.1-8B model, SyncKV's performance is nearly indistinguishable from the full attention baseline and surpasses all other methods. This remarkable efficiency is robustly maintained across different model families like Qwen and larger models, even at aggressive compression rates. The results unequivocally demonstrate the superiority of SyncKV. This success is largely attributed to SyncKV's ability to efficiently retrieve cyclically important KV cache, avoiding the permanent information loss inherent in static eviction methods.

**Information Retrieval Performance.** To evaluate the long-context information retrieval capabilities of the models, we designed and conducted a series of NIAH experiments. Figure 4 shows that SyncKV achieves a high score of 0.989. This performance is highly comparable to the original model, which strongly demonstrates that SyncKV can efficiently compress the KV cache while retaining critical information with extremely high fidelity, thereby ensuring both information integrity and reliability during long-context inference.

**Multi-Step Reasoning Performance.** To further assess the multi-step inference capabilities of SyncKV, we performed tests on three specific tasks from SCbench (Li et al., 2025): Semantic Retrieval, Global Information, and Multi-Tasking, while retaining the experimental configuration from the single-step evaluations. As shown in Table 2, the results indicate that SyncKV significantly outperforms all the baseline methods in semantic retrieval. In other tasks, its performance remains also highly comparable to that of the original model. This superior multi-step performance is primarily attributed to SyncKV's unique mechanism during the decoding phase, which effectively retrieves important tokens and prevents the loss of crucial historical information during long-range reasoning.

## 5.3 LATENCY

We evaluated the end-to-end latency of SyncKV against the baseline on the Llama-3.1-8B-Instruct model. For the prefill stage, we measure the Time To First Token (TTFT). For the decode stage, we calculate the average latency per token over 50 generated tokens.

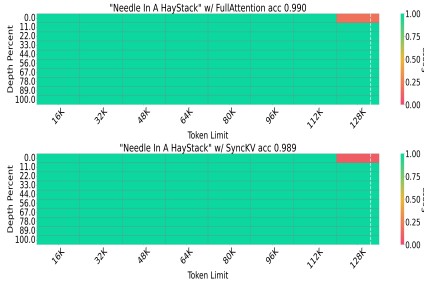
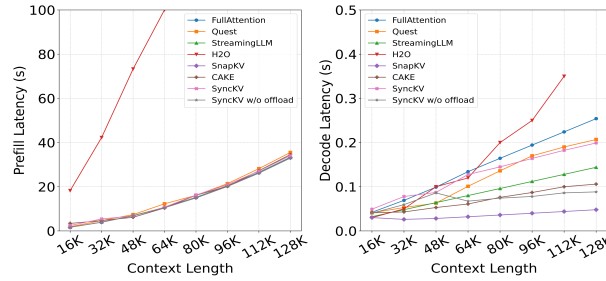

Figure 4: Results for Llama-3.1-8B-Instruct on NIAH, evaluated on context lengths from 16K to 128K tokens.

Figure 5: Comparison of end-to-end latency results between SyncKV and the baseline. The left and right plots show the latency for the prefill and decode stage, respectively.

The results are presented in Figure 5. During the prefill stage, SyncKV effectively hides the offload latency by overlapping it with the attention computation. In the decoding phase, while the CPU operations required by SyncKV's anticipation mechanism introduce additional latency, this is a deliberate trade-off necessary for the high-precision retrieval of crucial KV cache. Even so, it still achieves a 1.25x speedup over FullAttention at a context length of 128K tokens. Furthermore, we analyze SyncKV w/o offload, a variant that keeps all tensors on the GPU, assuming sufficient memory. Compared to Quest, SyncKV w/o offload achieves a comprehensive lead in both accuracy and speed under the same memory footprint. By eliminating the CPU operations associated with data uploading, this variant demonstrates the benefits of our computation model based on a sparse cache, achieving a 3x decoding speedup over FullAttention.

## 5.4 ABLATION STUDIES

We conducted a series of comprehensive ablation studies to validate the necessity of our core design choices and hyperparameter sensitivity in SyncKV. The results in Table 3 confirm the necessity of each component. Detailed experimental configurations and analysis are presented in Appendix D.

**w/o Initialization.** In this variant, we remove the KV cache offloading step during the prefill stage. The results show that while this reduces prefill latency by eliminating data transfer, it completely negates any GPU memory savings, which is the core problem our algorithm is designed to solve.

**w/o Anticipation.** This variant removes the dynamic update mechanism of the anticipation step, relying solely on a static token set. The results indicate that although this variant achieves the lowest decode latency due to its simplified computation, it suffers from a noticeable drop in task accuracy. This highlights the vital role of the anticipation step in maintaining reasoning capabilities.

**w/o Suspension.** This variant forces every decode step to be an anticipation step. Although this achieves the highest accuracy by most precisely tracking attention, it leads to a prohibitive increase in decode latency due to excessive computational and I/O overhead. This negates the acceleration benefits, proving that the suspension step is indispensable.

**Hyperparameter Sensitivity.** Furthermore, additional ablation studies, particularly those concerning hyperparameter sensitivity, are detailed in Appendix D. Studies justify our default hyperparameters ($\rho = 0.5$, $N_r = 2$, $m = 5$) as providing an optimal trade-off between performance and latency.

## 6 CONCLUSION

In this paper, we propose SyncKV, a training-free dynamic KV cache compression framework for efficient long-context LLM inference. Motivated by the attention drift phenomenon and insights into the temporal locality and intralayer similarity, SyncKV employs a syncopated strategy with representative heads and asynchronous offloading and reloading. Experiments show that SyncKV reduces KV cache memory usage by up to 80% without significant accuracy loss, and achieves state-of-the-art performance on multiple tasks. Our work offers an effective solution for deploying long-context LLMs on GPU memory-constrained hardware.

REPRODUCIBILITY STATEMENT

Hyperparameters, hardware environment, and other pertinent details are presented in Section 5. The core algorithm for SyncKV is provided in Section 4. Detailed settings and implementation of baselines can be found in Section 5 and Section C.

ETHICS STATEMENT

Our work introduces SyncKV, a framework for compressing the KV cache in Large Language Models to reduce its memory overhead. The research is purely algorithmic in nature, with a primary focus on memory optimization.

This study did not involve human subjects and all experiments were conducted on publicly available open-source models and standard academic benchmarks. No personal or private information was used. We do not foresee any direct ethical concerns arising from this work. We declare no conflict of interest.

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

## A  PSEUDOCODE

To provide a clear perspective on our proposed method, we present the detailed SyncKV workflow in the form of pseudocode. The entire step-by-step procedure is systematically illustrated in Algorithm 1.

---

**Algorithm 1** SyncKV Inference Process

---

1: **Input:** Prompt $P$, Model $M$, Stride $m$, Ratio $\rho$, Budget $k$, Clusters $N_r$, Layers $L$, Generation length $T$
2: **Output:** Generated sequence $Y$
3: *// Offline Head Clustering*
4: **for** $l = 1$ to $L$ **do**
5:     Compute similarity matrix $S^{(l)}$
6:     $\mathcal{H}_R^{(l)}, \mathcal{H}_{NR}^{(l)} \leftarrow$ K-Means$(S^{(l)}, N_r)$
7: **end for**
8: *// Prefill Stage*
9: **for** $l = 1$ to $L$ **do**
10:     $(K^{(l)}, V^{(l)}) \leftarrow$ GenerateKVForLayer$(M, P, l)$
11:     Partition $(K^{(l)}, V^{(l)})$ into GPU-resident $(K_R^{(l)}, V_R^{(l)})$ and offloadable $(K_{NR}^{(l)}, V_{NR}^{(l)})$
12:     Asynchronously transfer $(K_{NR}^{(l)}, V_{NR}^{(l)})$ to CPU memory
13:     **for** $g = 1$ to $N_r$ **do**
14:         $h_r^{(g)} \leftarrow$ Representative head in cluster $g$ of layer $l$
15:         $\mathcal{I}_0^{(l,g)} \leftarrow$ argTop$_k($Attention$(q_{L-1}, K^{(l,h_r^{(g)})}), k)$
16:         $\mathcal{I}_S^{(l,g)}, \mathcal{I}_{D,0}^{(l,g)} \leftarrow$ Split$(\mathcal{I}_0^{(l,g)}, \rho \cdot k)$
17:     **end for**
18: **end for**
19: *// Decode Stage*
20: **for** $t = 1$ to $T$ **do**
21:     **for** $l = 1$ to $L$ **do**
22:         $O_t^R \leftarrow$ Attention$_{\mathcal{H}_R^{(l)}}(q_t, K_R^{(l)}, V_R^{(l)})$
23:         $\mathcal{I}_{\text{sparse}} \leftarrow \bigcup_{g=1}^{N_r}\{\mathcal{I}_S^{(l,g)} \cup \mathcal{I}_{D,t-1}^{(l,g)}\}$
24:         $O_t^{NR} \leftarrow$ Attention$_{\mathcal{H}_{NR}^{(l)}}(q_t, K_{NR}^{(l)}[\mathcal{I}_{\text{sparse}}], V_{NR}^{(l)}[\mathcal{I}_{\text{sparse}}])$
25:         $O_t \leftarrow$ Combine$(O_t^R, O_t^{NR})$
26:         **if** $t \pmod{m} = 0$ **then**
27:             **for** $g = 1$ to $N_r$ **do**
28:                 $A_t^{(g)} \leftarrow$ Attention$(q_t, K^{(l,h_r^{(g)})})$
29:                 $\mathcal{I}_{D,t}^{(l,g)} \leftarrow$ argTop$_{k_{dyn}}(A_t^{(g)}, k_{dyn})$
30:                 Fetch KV for $\mathcal{I}_{D,t}^{(l,g)}$ from CPU
31:             **end for**
32:         **else**
33:             **for** $g = 1$ to $N_r$ **do**
34:                 $\mathcal{I}_{D,t}^{(l,g)} \leftarrow \mathcal{I}_{D,t-1}^{(l,g)}$
35:             **end for**
36:         **end if**
37:     **end for**
38:     $y_t, (K, V) \leftarrow$ DecodeForward$(O_t, K, V)$
39:     Append $y_t$ to $Y$
40: **end for**
41: **return** $Y$

---

## B    IMPLEMENTATION DETAILS

### B.1    HEADS CLUSTER

**Robustness Analysis.** To validate the stability and generalizability of our clustering methodology, we first performed a robustness analysis. Specifically, we constructed multiple separate and non-overlapping corpora to test the consistency of our results. Each corpus was created by randomly sampling 50 unique articles from a comprehensive snapshot of English Wikipedia. To ensure fair comparison and maintain methodological consistency, every article across all corpora was truncated to its first 1000 tokens. Despite the textual variations across these independently sampled corpora, the resulting head cluster configurations exhibited strong concordance when the clustering process was applied to each. The cluster assignments for the vast majority of heads remained consistent across these experiments, indicating that the functional specializations we identified are not artifacts of a specific text sample but rather intrinsic properties of the model's attention mechanism.

**Handling of MHA and GQA Architectures.** Our method for calculating the overlap coefficient is adapted for different architectures of attention. For the standard Multi-Head Attention (MHA) architecture, we directly calculate the overlap coefficient between any two individual heads, $h_i$ and $h_j$. In contrast, for the Grouped-Query Attention (GQA) architecture, query heads within the same group are functionally coupled, as they share a common set of keys and values. Therefore, before computing coefficients, we first apply a max-pooling operation across the attention head dimension within each group. This step produces a single consolidated attention map that represents the collective function of the group. Subsequent overlap coefficient analysis is then performed on these pooled, group-level representations to measure the affinity between different functional groups.

Our offline head clustering process consists of the following core steps:

1. **Corpus Construction & Weight Extraction**: We first construct a representative corpus from 50 English Wikipedia articles, each truncated to 1000 tokens. We then feed this text into an LLM to extract and save the complete attention weight matrices $A_t^{(l,h)}$ for each layer at each decode step.

2. **Similarity Matrix Calculation**: Based on the extracted weights, we calculate a head-to-head similarity matrix $S^{(l)}$ for each layer $l$. The similarity between two heads, $h_i$ and $h_j$, is defined as the average overlap of their top $k$ token index sets, $\mathcal{I}^{(l,h)}$, attended within a local window.

3. **Distance Conversion & Clustering**: We convert the similarity matrix $S^{(l)}$ into a distance matrix $D^{(l)}$ via the formula $D_{i,j}^{(l)} = \max(S^{(l)}) - S_{i,j}^{(l)}$. This distance matrix serves as the input feature for the K-Means algorithm, which we then apply to partition all effective heads into $N_r$ groups, $\mathcal{C}_g$.

4. **R-Heads Selection**: Finally, for each resulting cluster $\mathcal{C}_g$, we select the head closest to the cluster's centroid to be its R-Head, which is considered the most functionally representative member.

This offline process ensures that SyncKV incurs no computational overhead from clustering during actual inference, thus guaranteeing its efficiency.

## C    EXPERIMENT DETAILS

### C.1    BASELINE METHODS.

We evaluated SyncKV performance against many baselines: 1) Full Attention, the original model, does not evict any tokens. 2) Quest. A query-aware selection method that speeds up attention by dynamically choosing KV pages, but retains the full cache and does not save memory. For Quest, we follow the settings in its paper, setting the token budget to 2048 and the size of chunk to 16. 3) StreamingLLM. It uses an initial window and a sliding window to retain tokens, which means that the tokens in the middle are significantly evicted. 4) H2O. It evicts tokens based on cumulative attention scores from the prefill and decode stages. 5) SnapKV. It remains the token with the top $k$ window attention aggregation in the prefill stage. 6) CAKE. Adaptively allocates cache size per layer based on attention dynamics and evicts tokens using an indicator of their importance and variability.

Table 4: The performance of Llama-3.1-8B-Instruct and Qwen2.5-7B-Instruct on LongBench.

| Methods | Mem% | Avg. | Single-Doc QA | Multi-Doc QA | Summarize | Few-Shot | Synthetic | Code |
|---|---|---|---|---|---|---|---|---|
| *Llama-3.1-8B-Instruct* | 100% | 49.76 | 43.39 | 42.76 | 29.20 | 69.39 | 51.81 | 62.02 |
| Quest | 100% | 46.86 | 42.05 | 37.38 | **28.92** | 64.81 | **52.98** | 55.03 |
| StreamingLLM | 30% | 41.65 | 30.73 | 35.26 | 24.72 | 66.20 | 36.84 | 56.14 |
| H2O | 30% | 39.92 | 36.00 | 37.07 | 26.67 | 66.36 | 27.84 | 45.56 |
| SnapKV | 30% | 43.67 | 36.11 | 37.72 | 23.70 | 67.88 | 48.12 | 48.46 |
| CAKE | 30% | 47.63 | 42.23 | 40.52 | 26.59 | **68.58** | 50.88 | 56.97 |
| SyncKV | 30% | **48.84** | **42.82** | **41.98** | 26.89 | 68.50 | 52.30 | **60.55** |
| *Qwen2.5-7B-Instruct* | 100% | 49.70 | 41.59 | 43.96 | 26.72 | 67.56 | 54.75 | 63.63 |
| Quest | 100% | 46.66 | 40.14 | 41.73 | 22.95 | 67.75 | 51.25 | 56.13 |
| StreamingLLM | 50% | 43.42 | 32.95 | 39.39 | 24.30 | 66.50 | 37.00 | 60.40 |
| H2O | 50% | 41.66 | 33.93 | 37.33 | 23.39 | 61.81 | 47.00 | 46.52 |
| SnapKV | 50% | 45.14 | 36.33 | 41.85 | 21.56 | 65.32 | **54.75** | 51.05 |
| CAKE | 30% | 49.31 | **41.29** | 43.92 | 24.58 | 68.00 | 54.25 | **63.85** |
| SyncKV | 50% | **49.55** | 40.86 | **44.84** | **24.91** | **68.79** | 54.25 | 63.63 |

## C.2 Evaluating Tasks.

To evaluate the performance of SyncKV and other baselines, we use three designed benchmarks:
(1) LongBench (Bai et al., 2023): This benchmark focuses on evaluating the understanding and
reasoning capabilities of LLMs in single-step reasoning. We set the maximum context length to
128K tokens. (2) SCBench (Li et al., 2025): This benchmark is designed to evaluate a model's
multi-step reasoning performance. Given that the average input length in this benchmark ranges
from 22K to 1.5M tokens, we standardized our evaluation by truncating all input sequences to 128K
tokens. (3) Needle In A Haystack (Fu et al., 2024): This test evaluates the in-context retrieval
capabilities of LLMs. It measures retrieval accuracy under high distraction and diagnoses potential
positional biases by embedding a target "needle" of information into a long "haystack" of text. we
covered a range of context lengths from 16K to 128K tokens, with tests conducted at increasing
intervals of 16K.

## C.3 LongBench Results

In this section, we present the performance of Llama-3.1-8B-Instruct and Qwen2.5-7B-Instruct on
LongBench. As shown in Table 4, SyncKV still demonstrates a performance far exceeding that of
other compression algorithms.

# D Ablation Studies

To systematically validate the necessity of each component in our SyncKV design and to provide a
rationale for our hyperparameter choices, we conducted a series of detailed ablation studies. This
section elaborates on these experiments, for which we reported the impact on accuracy and latency
on the Qasper dataset from LongBench, with the latency metric tested over an average context length
of 16K.

We analyze the hyperparameters that impact the performance and efficiency of SyncKV to determine
their optimal values.

**Static Ratio $\rho$.** The static ratio $\rho$, controls the proportion of initial $\mathcal{I}_0^{(g)}$ that are permanently re-
tained on the GPU. We tested multiple values for $\rho$ in the range [0,1]. As shown in Figure 6, the
results indicate that when $\rho$ is too low, the model retains an insufficient global context, which affects
tasks that require an understanding of the overall theme. Conversely, when $\rho$ is too high, the budget
reserved for dynamic information is inadequate, weakening the model's ability to adapt to "atten-
tion drift." Ultimately, we found that $\rho = 0.5$ strikes the optimal balance between preserving global
context and adapting to dynamic attention shifts.

**Number of R-Heads $N_r$.** The number of R-Heads, $N_r$, determines the granularity of the cluster-
ing. As shown in Table 3, our experimental results indicate that the setting $N_r = 2$ achieves optimal

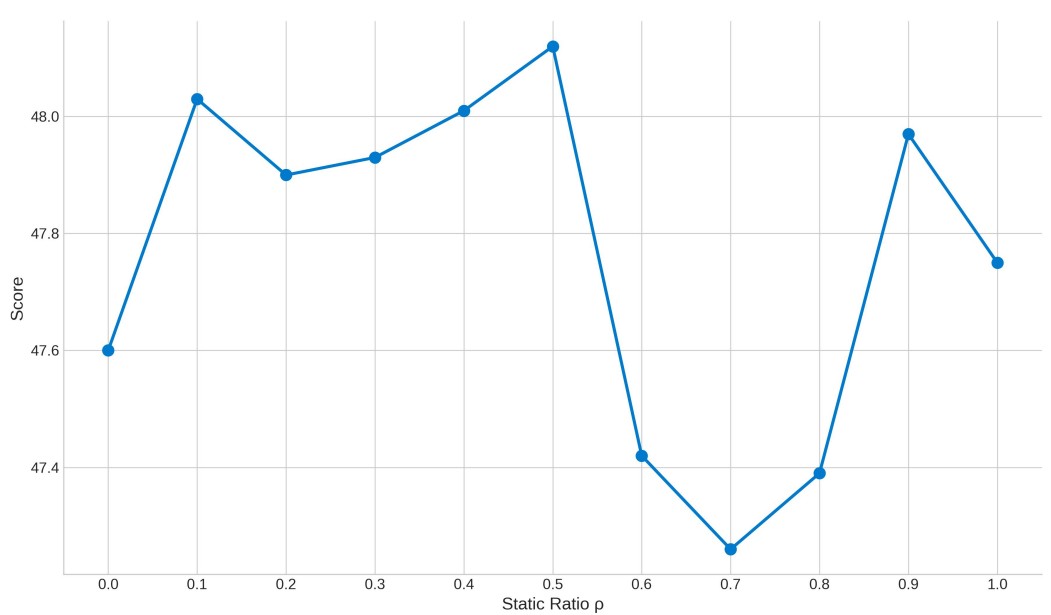

Figure 6: The impact of the Static Ratio $\rho$ on model performance.

performance. We attribute this to a critical trade-off between functional specialization and budget allocation. An overly small $N_r$ fails to account for the functional heterogeneity among attention heads, treating them as a monolithic group. This overlooks their specialized roles and can lead to less precise retention of the key-value cache. In contrast, a large $N_r$ fragments the limited token budget too thinly in many clusters. This provides each NR-Head with insufficient contextual information to make effective decisions, ultimately degrading model performance. Therefore, setting $N_r = 2$ strikes an effective balance, allowing sufficient differentiation of head functionalities without overly constraining the token budget for each group. This result strongly supports our choice of $N_r = 2$ as the optimal trade-off between model performance and resource constraints.

**Synchronization Stride $m$.** The synchronization stride $m$, defines the frequency of the anticipation stage. When $m = 1$, it is equivalent to the "w/o Suspension" case, resulting in the highest latency. As $m$ increases, token generation throughput improves significantly. However, if m becomes too large, the dynamic information in the cache becomes stale, leading to a decrease in model performance. As shown in Table 3, our experiments show that $m = 5$ offers an optimal compromise between inference latency and model performance.

# E  STATEMENT ON THE USE OF LARGE LANGUAGE MODELS

We report on the use of Large Language Models (LLMs) in the preparation of this paper. The use of LLMs was strictly limited to the role of a general-purpose writing assistant. Specifically, we used these tools to proofread, correct grammatical errors, and rephrase sentences to improve clarity and readability. The LLMs did not contribute to the core scientific aspects of this work, such as research ideation, experimental design, data analysis, or the generation of substantive content.

