# OpenReview forum: "SyncKV: A Syncopated Scheduling Approach to KV Cache Compression for Efficient Long-Context LLM Inference"
_ICLR.cc/2026/Conference — ICLR 2026 Conference Withdrawn Submission_

### Official Review · Reviewer_xT4B · 2025-10-19

**Soundness:** 2
**Presentation:** 3
**Contribution:** 2
**Rating:** 4
**Confidence:** 4

**Summary:**

This paper introduces SyncKV, a training-free dynamic KV cache compression framework designed to mitigate the significant GPU memory overhead associated with long-context LLM inference. The core problem addressed is that existing static compression methods, which evict tokens based on initial importance scores, fail to account for "attention drift"—the phenomenon where token importance changes dynamically throughout the generation process. The authors' analysis shows that the overlap of important tokens between the prefill stage and later generation steps can drop to as low as 30% after just 50 tokens.
SyncKV is built upon two key insights: (1) the set of important tokens exhibits high temporal locality, and (2) there is high intralayer similarity among attention heads within the same layer. The authors claim this approach reduces KV cache memory usage by up to 80% while achieving state-of-the-art performance on multiple long-context benchmarks.

**Strengths:**

* The observation of "Temporal Locality" is insightful and potentially effective.
* The ablation studies are relatively sufficient and comprehensive.

**Weaknesses:**

* The concept of "attention drift" is not entirely new [1,2]. Please compare with these in the related work section to clarify the contribution of this work.
* The current method's utilization of Locality seems somewhat simplistic. Perhaps the Anticipation Step should be triggered dynamically based on the input.
* On SCBench, the performance did not surpass CAKE. Could you explore the impact of adjusting `m` on the performance of multi-step reasoning tasks? I suspect that most tasks in LongBench, except for summarization, are not very sensitive to `m`.

[1] InfiniGen: Efficient Generative Inference of Large Language Models with Dynamic KV Cache Management (OSDI 24)

[2] OmniKV: Dynamic Context Selection for Efficient Long-Context LLMs (ICLR 25)

**Questions:**

* As shown in Figure 2b, there are some decoding steps that have a low overlap with all previous steps. This might indicate a difficult or pivotal token. If you do not perform an Anticipation Step at this point, would it lead to a degradation in performance?
* Could you test the performance of SyncKV on a reasoning model such as DeepSeek-R1-Distill-Llama-8B?

---

### Official Review · Reviewer_rt7o · 2025-10-31

**Soundness:** 2
**Presentation:** 3
**Contribution:** 3
**Rating:** 4
**Confidence:** 3

**Summary:**

This paper proposes **SyncKV**, a training-free dynamic KV cache compression framework for long-context LLM inference. Based on key observations — attention drift, temporal locality, and intralayer similarity — SyncKV employs representative heads to periodically update important tokens and asynchronously transfer KV cache between CPU and GPU, significantly reducing memory usage without notable accuracy loss. Experiments on multiple open-source LLMs show state-of-the-art accuracy–efficiency trade-offs across long-context benchmarks.

**Strengths:**

1. Novel Observations: Identifies and analyzes key phenomena in long-context attention, including attention drift, temporal locality, and intralayer similarity, which inform the design of SyncKV.
2. Methodological Contribution: Proposes a syncopated scheduling strategy with representative heads and asynchronous KV cache transfer, enabling dynamic retrieval without retraining.
3. Empirical Performance: Demonstrates state-of-the-art accuracy–efficiency trade-offs and up to 80% GPU KV cache memory reduction across multiple open-source LLMs and benchmarks.

**Weaknesses:**

1. It is unclear whether attention drift occurs uniformly across all heads and how fixed-pattern heads might be more efficiently handled.
2. The necessity and rationale of head clustering, as well as its design choices (e.g., cluster count), are not sufficiently justified or empirically validated.
3. SyncKV provides only modest accuracy gains over CAKE while incurring higher latency, raising concerns about its practical advantage in certain scenarios.

**Questions:**

1. Is attention drift observed consistently across all heads? Prior works ([1][2][3]) have categorized heads into fixed-pattern and dynamically changing types. If some heads follow fixed patterns, could they omit dynamic cache updates or adopt a larger synchronization stride $m$?
2. Could the authors clarify the necessity of head clustering? Specifically, please explain the rationale behind this design, including whether the heads exhibit the same similarity across different decoding steps. In addition, provide evidence of the performance benefits it brings and clarify how the number of clusters was determined.
3. Comparing Table 1 and Figure 5, SyncKV shows only modest accuracy improvements over CAKE but higher latency. Does this imply it may not be optimal in many scenarios? To better highlight SyncKV’s accuracy advantage, could additional benchmarks, such as RULER, be included for a more comprehensive evaluation?

If the authors can satisfactorily address these concerns and provide additional evidence, I would be inclined to raise my rating.

[1] RazorAttention: Efficient KV Cache Compression Through Retrieval Heads

[2] DuoAttention: Efficient Long-Context LLM Inference with Retrieval and Streaming Heads

[3] CateKV: On Sequential Consistency for Long-Context LLM Inference Acceleration

---

### Official Review · Reviewer_o7Ke · 2025-11-02

**Soundness:** 2
**Presentation:** 3
**Contribution:** 2
**Rating:** 2
**Confidence:** 4

**Summary:**

This paper proposes SyncKV, a dynamic KV cache offloading method that addresses the "attention drift" phenomenon in long-context LLM inference. The approach employs representative heads to identify important tokens and offload/reload KV cache between CPU and GPU memory.

**Strengths:**

1. The integration of sparse attention mechanisms with offloading represents a valuable research direction for addressing long-context inference challenges.
2. The paper makes a valuable observation by identifying and quantifying the attention drift phenomenon.

**Weaknesses:**

1. The paper fails to compare against existing sparse attention methods with offloading capabilities, such as InfLLM and ShadowKV, which are directly relevant to the proposed approach.

2. The primary benchmark used (LongBench-v1) has an average sequence length of only 8K tokens, which is insufficient to validate claims about long-context performance. The paper would benefit significantly from evaluation on more challenging benchmarks designed for longer contexts, such as RULER or LongBench-v2.

3. The experimental results show only minimal average performance improvements over existing methods like QUEST and CAKE, while incurring substantially higher inference latency. This unfavorable accuracy-latency trade-off undermines the practical value of the proposed method.

**Questions:**

Please refer to Weaknesses.

---

### Note · Authors · 2025-12-01

**Comment:**

We sincerely thank the reviewers for their constructive comments. We have decided to withdraw our submission to incorporate these suggestions and further improve the paper for future work.

**Withdrawal Confirmation:**

I have read and agree with the venue's withdrawal policy on behalf of myself and my co-authors.